# Comparative Effectiveness of Two Models of Point-of-Care Ultrasound for Detection of Post-Void Residual Urine during Acute Ischemic Stroke: Preliminary Findings of Real-World Clinical Application

**DOI:** 10.3390/diagnostics13152599

**Published:** 2023-08-04

**Authors:** Wan-Ling Chang, Shu-Hui Lai, Chu-Fang Cheng, Valeria Chiu, Shinn-Kuang Lin

**Affiliations:** 1Stroke Center and Department of Neurology, Taipei Tzu Chi Hospital, Buddhist Tzu Chi Medical Foundation, New Taipei City 23142, Taiwan; laetitia0717@hotmail.com (W.-L.C.); yamato21388@gmail.com (C.-F.C.); 2Department of Nursing, Taipei Tzu Chi Hospital, Buddhist Tzu Chi Medical Foundation, New Taipei City 23142, Taiwan; shlgreatvision@gmail.com; 3Department of Physical Medicine and Rehabilitation, Taipei Tzu Chi Hospital, Buddhist Tzu Chi Medical Foundation, New Taipei City 23142, Taiwan; haydenbell28@gmail.com; 4School of Medicine, Tzu Chi University, Hualien 97004, Taiwan

**Keywords:** acute ischemic stroke, bladder ultrasound, point-of-care ultrasound, post-void residual urine

## Abstract

We conducted a comparative study of two models of point-of-care ultrasound devices for measuring post-void residual urine (PVRU). We prospectively enrolled 55 stroke inpatients who underwent both real-time B-mode ultrasound (Device A) and automated three-dimensional (3D) scanning ultrasound (Device B), with a total of 108 measurements. The median PVRU volume of Device B was 40 mL larger than that of Device A. The PVRU difference between the devices was positively and linearly correlated with PVRU. The correlation of PVRU volume between the devices was strong, but the agreement level was only moderate. Measurement deviations were observed in 43 (40%) and 11 (10%) measurements with Device B and Device A, respectively. The PVRU volume was low in spherical bladder shapes but sequentially increased in triangular, undefined, ellipsoid, and cuboid bladder shapes. Further comparison of 60 sets of PVRU without measurement deviations revealed higher agreements between the devices at correction coefficients of 0.52, 0.66, and 0.81 for PVRU volumes of <100, 100–200, and >200 mL, respectively. The automated 3D scanning ultrasound is more convenient for learning and scanning, but it exhibits larger measurement deviations. Real-time B-mode ultrasound accurately visualizes the urinary bladder but tends to underestimate the urinary bladder when the PVRU volume is large. Hence, real-time B-mode ultrasound with automated PVRU-based adjustment of calculation formulas may be a better solution for estimating bladder volume.

## 1. Introduction

Urinary tract infection (UTI) is one of the most common complications in stroke patients. UTI may present either as an asymptomatic disorder, which is often neglected during hospitalization, or as a serious infection that necessitates an extended hospital stay or poses a threat to the lives of immunocompromised patients or patients with multiple comorbidities. Urinary complications have been documented to be an independent factor of prolonged hospital stay [1]. Factors associated with UTI after an acute stroke include direct brain injury to the central micturition pathway, which may result in detrusor areflexia during the cerebral shock stage [2], or detrusor external sphincter dyssynergia [3], lower urinary tract dysfunction, premorbid dysuria due to diabetic cystopathy, benign prostate hypertrophy, or other causes of neurogenic bladder, prolonged urinary catheterization, and inadequate local hygiene [4]. Although urinary incontinence occurs in 46% to 60% of stroke inpatients [5], urinary retention, which has been reported in 29% to 47% of stroke inpatients [3,6], is the most common cause of UTI.

With the advancement of ultrasound technology, high-resolution, cost-effective, and portable ultrasound devices have opened up new avenues for the clinical application of point-of-care ultrasound (POCUS). This technology has enabled ultrasound examinations to be conducted at the patient’s location by the health-care provider in real time, thus aiding in the clinical decision-making process [7]. Bladder ultrasound is the most effective and convenient method for measuring the post-void residual urine (PVRU) volume and detecting lower urinary tract dysfunction, specifically urinary retention [8]. Because of its potential to reduce the incidence of UTI and shorten hospital length of stay, bladder POCUS has been recommended for PVRU measurement as a routine procedure for stroke inpatients who are at an increased risk of developing UTI [4].

In a previous study, we established AGN3 criteria for selecting inpatients with a high risk of developing UTI following an acute stroke [4]. These AGN3 criteria comprise five items pertaining to clinical features, namely age ≥ 75 years, female sex, an initial total National Institutes of Health Stroke Scale (NIHSS) score of ≥5, an initial NIHSS conscious score of ≥1, and an initial NIHSS leg score of ≥2. Patients who are admitted to a stroke ward following an acute stroke and who meet one or more of these AGN3 criteria are deemed eligible for bladder POCUS to determine their PVRU volume. Currently, bladder POCUS is a routine medical procedure at our stroke ward for stroke patients during their first week of hospitalization and is implemented to prevent UTI.

Two models of POCUS devices are available at our stroke ward. The first model is a wireless hand-held device that outputs real-time B-mode images. This model required bladder visualization followed by measurement of the diameter of the urinary bladder to determine the PVRU volume. The second model is also a hand-held device but does not output real-time B-mode images. This model enables the rapid determination of the PVRU volume through automated scanning without requiring urinary bladder visualization (henceforth referred to as the “blind method”). However, certain discrepancies in the PVRU volume measured by these two devices have been observed. To better understand the effectiveness and convenience of these two models of POCUS devices in measuring PVRU volume, we conducted a comparative study.

## 2. Materials and Methods

### 2.1. Design and Participants

This prospective study was conducted in a neurological stroke ward between August 2022 and February 2023. The study was conducted in accordance with the ethical principles of the Declaration of Helsinki, and the study protocol was approved by the Institutional Review Board of Taipei Tzu Chi Hospital, Buddhist Tzu Chi Medical Foundation (approval number: 11X-025). Each item in the AGN3 criteria was assigned 1 point, with the total AGN3 score ranging from 0 to 5 [4]. Patients who received a diagnosis of acute ischemic stroke and were admitted to the stroke ward with an AGN3 score of ≥1 were included in the study. All patients provided written informed consent. Patients who had a urinary catheter inserted at the emergency department before their admission to the ward were excluded.

### 2.2. Instruments and Measurements

In accordance with the protocol of bladder POCUS in our stroke ward, PVRU volume was measured twice on different days within 1 week of hospitalization. Bladder ultrasound scanning was performed in a supine position within 15 min of urinary voiding with the two POCUS devices. To achieve convenient bedside application, the two devices were placed on the same stand, which was equipped with wheels. Initially, the ultrasound operators used a wireless pocket-sized ASUS LU700C portable ultrasound scanner (Device A; ASUSTek Computer, Taipei, Taiwan) with a 2–5-MHz transducer capable of two-dimensional (2D) real-time color B-mode imaging. Sonographic images were then obtained by the scanner, transferred over Wi-Fi, and displayed on a tablet. A split-image mode was used to simultaneously depict the horizontal and vertical planes of the urinary bladder images on the screen. Subsequently, the ultrasound operators manually measured the width and depth of the bladder on the horizontal plane and the height of the bladder on the vertical plane. The PVRU volume was then automatically calculated using the following built-in formula and displayed in units of milliliters: width × depth × height × 0.52 [9,10,11]. Immediately after the first measurement, the operators proceeded to measure the PVRU volume with a Kaixin BVT01 portable bladder scanner (Device B; Xuzhou Kaixin Electronic Instrument, Xuzhou, China). They placed Device B on the lower abdomen just above the pubic symphysis, with the probe attached to the patient’s skin. The device provided an automated three-dimensional (3D) scan of the urinary bladder and displayed the automatically calculated PVRU volume through a built-in algorithm. Twelve 2D images demonstrating the urinary bladder with automated bladder outline-tracking in six sections were displayed on the screen on six pages. Manual adjustment of outline tracking was achieved with a pencil tool for each image to correct tracking errors. After the scanning results were saved, the scanned images captured by the two devices were retrieved on a personal computer.

Bladder POCUS was performed by seven rotational in-charge resident physicians and two long-term care advanced practice registered nurses. Before measurement, all ultrasound operators underwent a short-term training program, including a video detailing the procedure, to teach them how to measure the PVRU volume with the two different ultrasound devices. Following bladder POCUS scanning, the drained urine volume was recorded from a subsequent urinary catheterization procedure within 30 min in cases of large PVRU volume.

All ultrasound images with measured PVRU volumes were stored on the devices for a final review by a stroke neurologist specialized in neurosonology with over 30 years of clinical experience. The bladder shapes evaluated with Device A were classified as spherical, triangular, ellipsoid, cuboid, and undefined bladder (Figure 1) [12]. Two definitions of measurement deviations were used. Type I deviation was used to refer to a clear deviation in either the location or distance of the measuring lines of the bladder diameter (Device A) or a clear deviation in the automated edge tracking of the bladder wall during 3D measurement (Device B). Type II deviation was used to refer to an incorrect measurement of the distance or automated edge tracking of a tissue that did not belong to the urinary bladder.

After all ultrasound measurements, each ultrasound operator completed a questionnaire on the learning times, measurement times, and satisfaction level of the two ultrasound devices. Each operator subsequently provided a score, ranging from 0 to 10, on the convenience of the two devices, with the higher scores indicating higher levels of satisfaction.

### 2.3. Statistical Analyses

Because the measured variables had a skewed distribution, medians alongside the 25th and 75th percentiles were used to explain the range. The Mann–Whitney *U* test was used to evaluate differences in continuous variables. The Wilcoxon test was used to compare the measured PVRU volumes between the two devices. Linear regression analysis was used to evaluate the correlation of the difference in measured PVRU volume between the two devices. A *p* value less than 0.05 was used to indicate statistical significance. Spearman’s rank correlation coefficient, intraclass correlation coefficient (ICC), and concordance correlation coefficient (CCC) were used to evaluate the agreement, correlation, and reliability, respectively, between the PVRU volumes measured with the two ultrasound devices. The ICC and CCC values ranged from 0 to 1, with values closer to 1 indicating a greater level of homogeneity. All statistical analyses were conducted using MedCalc version 18 (MedCalc Software bvbd, Ostend, Belgium).

## 3. Results

### 3.1. Participant Characteristics

A total of 55 patients (23 men and 32 women, median age: 74 (65–85) years) who met the inclusion criteria were included in the study. A total of 108 PVRU measurements with Devices A and B were conducted. PVRU measurements were conducted once in 4 patients, twice in 49 patients, and three times in 2 patients. Table 1 summarizes the basic characteristics of the participants. Although the men were older and had lower AGN3 scores than did the women, their results did not reach statistical significance. Compared with women, men had a higher body mass index (BMI). Small artery occlusion comprised the majority of the stroke subtypes, followed by large artery atherosclerosis, cardioembolism, and other determined etiology. The numbers of cerebral infarction involving the cortical or subcortical area of the middle cerebral arterial territory, the supratentorial deep brain structures (including basal ganglia and thalamus), the brainstem, and the cerebellum were 21, 20, 11, and 3, respectively.

### 3.2. Bladder POCUS Findings

Table 2 presents the results of PVRU measurements by bladder POCUS. In both men and women, the median PVRU volume measured with Device B (105 mL) was larger than that measured with Device A (65 mL). Regardless of whether Device A or B was used, the PVRU volumes were higher in men than in women. Spearman’s rank correlation coefficient analysis revealed a positive correlation between age and PVRU volume with both Device A (ρ = 0.362, *p* < 0.001) and Device B (ρ = 0.235, *p* = 0.014).

A PVRU volume of >100 mL was observed in 37% and 52% of measurements with Device A and Device B, respectively. Regardless of whether Device A or B was used, the rate of PVRU volume >100 mL was higher in men than in women. The median difference in PVRU volume between the devices was 40 (10–95) mL. The difference in measured PVRU volume between the devices was a positively and linearly correlated with measured PVRU volume for both Device A (*r^2^* = 0.14, *p* < 0.001) and Device B (*r^2^* = 0.67, *p* < 0.001) (Figure 2). Differences in PVRU of ≥50 mL and of ≥100 mL were observed in 47 (44%) and 23 (21%) measurements, respectively. Spearman’s rank correlation coefficient analysis revealed a strong positive, linear correlation between the devices (*r* = 0.873, *p* < 0.001; Figure 3A). Agreement analysis only revealed a moderate level of agreement between the devices in ICC (ICC = 0.688, Figure 3B) and in CCC (ρ_c_ = 0.686, Figure 3C) analysis.

### 3.3. Comparison of PVRU Volumes Measured with POCUS and Catheterization

Only six patients underwent urinary catheterization for urine drainage after POCUS measurement of PVRU volume. In these six patients, the amount of drained urine by catheterization ranged from 300 to 500 mL (average = 408 mL; median = 425 mL). The amounts of PVRU measured with Device A and Device B ranged from 221 to 345 mL (average = 273 mL; median = 262 mL) and from 280 to 537 mL (average = 379 mL; median = 374 mL), respectively. The amounts of urine measured with urinary catheterization were larger than those of POCUS-measured PVRU. The average differences between the volume of POCUS-measured PVRU with Devices A and B and the volume of urine measured with urinary catheterization were 151 mL (35%) and 109 mL (27%), respectively.

### 3.4. Measurement Deviations of Bladder POCUS

Table 3 presents the measurement deviations of Devices A and B. A total of 43 measurement deviations (40%) were observed with Device B, which is much higher compared with the 11 measurement deviations (10%) observed with Device A (*p* < 0.001). Figure 4 presents the measurement deviations of the two devices. Although the presence of measurement deviations did not strongly correlate with age, gender, BMI, or PVRU volumes, such deviations tended to occur in older patients with larger amounts of PVRU measured with Device A. All measurement deviations observed with Device A were of Type I. By contrast, 30 Type I deviations (28%) and 13 Type II deviations (12%) were observed with Device B. Figure 3 depicts the two types of measurement deviations observed with the two devices.

### 3.5. Correlation between Bladder Shape and PVRU Volume

Table 4 presents the correlation between bladder shape and PVRU volumes with measurement deviations. Spherical and triangular bladders constituted 52% of all bladders. Only 10% of all bladders had an undefined shape. Regardless of whether Device A or B was used, the PVRU volumes were low in spherical bladders and sequentially increased in triangular, undefined, ellipsoid, and cuboid bladder shapes. When Device A was used, the number of measurement deviations considerably increased for ellipsoid and undefined bladder shapes. However, when Device B was used, no difference was observed in measurement deviations among bladder shapes. Spherical bladder shape was observed more frequently in women with smaller amounts of PVRU than in those with larger amounts of PVRU. Large amounts of PVRU were observed in men with ellipsoid bladder shape and in women with cuboid bladder shape.

To examine the correlation of PVRU volumes between Devices A and B, we excluded all the samples with measurement deviations, resulting in 60 sets of PVRU measurements without measurement deviations. For Device A, we calculated the PVRU volumes with different correction coefficients, namely 0.52, 0.66, 0.72, and 0.81. We then stratified these PVRU volumes into three subgroups, as follows, depending on the results of Device B: <100 mL, 100–200 mL, and >200 mL. Table 5 presents the results of agreement analysis results of ICC and CCC. For all 60 sets of measurements, the strongest agreement was observed with a coefficient of 0.72 (ICC = 0.902, CCC = 0.901). When stratification was performed for different PVRU volumes, the strongest agreements were observed with coefficients of 0.52 (ICC = 0.545, CCC = 0.537), 0.66 (ICC = 0.691, CCC = 0.671), and 0.81 (ICC = 0.591, CCC = 0.576), for PVRU volumes of <100 mL, 100–200 mL, and >200 mL, respectively.

### 3.6. Questionnaire Results

A total of nine resident physicians and advanced practice registered nurses conducted bladder POCUS measurements as follows. Three operators conducted bladder POCUS measurements more than 15 times, two operators conducted bladder POCUS measurements 6–10 times, and four operators conducted bladder POCUS measurements 5 times or fewer. Figure 5 presents the results of the questionnaire after all ultrasound operators completed their POCUS examinations. The time required to learn the bladder POCUS procedure was ≤5 min for seven operators (78%) with Device B and one operator (11%) with Device A. All bedside measurements of PVRU volume were completed within 10 min. The average time required to complete the bladder POCUS measurement was ≤5 min for five operators (56%) with Device A and seven operators (78%) with Device B. In terms of the convenience of bladder POCUS, eight operators provided scores ranging between 7 and 8 points for Device A, with an average of 7.8 points among all operators. In addition, seven operators provided scores ranging between 9 and 10 points for Device B, with an average score of 8.9 points among all operators.

## 4. Discussion

In this study, we investigated the performance of two models of bladder POCUS devices (Device A and Device B) for the rapid bedside measurement of PVRU volume in stroke inpatients. Device A (2D real-time B-mode ultrasound) provided direct visualization of the urinary bladder, thereby enabling accurate measurement of bladder diameters with few measurement deviations. Device B (automated 3D scanning) provided a more convenient and rapid measurement of bladder volumes with a shorter learning time but resulted in a larger number of measurement deviations. The median PVRU volume measured with Device B was 40 mL larger than that measured with Device A. When the PVRU volume increased, the difference between the devices became more apparent. Although a strong correlation of measured PVRU volume was observed between the devices, the agreement level between the devices was only moderate. The volume of urine measured with urinary catheterization was larger than that of PVRU measured with the two devices, with the volume of PVRU measured with Device B being closer to the volume of urine measured with urinary catheterization. Adjustment of the correction coefficients in accordance with the PVRU volumes for Device A improved the agreement between Devices A and B.

Urinary catheterization is the gold standard for measuring the volume of PVRU, although certain volume variabilities may be observed after catheterization. Because of the discomforting, time-consuming, and relatively invasive nature of this procedure, together with its unexpected complications, such as infection or trauma, it is not considered the primary method for PVRU evaluation. Bladder ultrasound is currently regarded as the most appropriate noninvasive first-line method for PVRU evaluation. According to a meta-analysis conducted by Palse et al. [13], the use of an ultrasound bladder scanner for PVRU evaluation and monitoring in immediate postoperative patients prevented unnecessary catheterizations and the risk of UTI.

In conventional ultrasound devices used for measuring the volume of PVRU, B-mode imaging is primarily used to directly visualize the urinary bladder through transverse and sagittal views. Dicuio et al. [10] compared the efficacy of five methods for calculating bladder volume: the prolate ellipsoid method, the double-area method, the double-ellipsoid method, the one-dimension method of bladder shape outlined manually with the maximum longitudinal diameter, and the one-dimension method of bladder shape outlined by a smooth ellipsoid with the maximum longitudinal diameter. They discovered all methods had the same precision, with errors not exceeding 25% of the voided volume. Hvarness et al. [14] compared three ultrasound calculation methods and recommended the use of the prolate ellipsoid method as the standard calculation method because of its simplicity. With the prolate ellipsoid method, after the width, depth, and height of the bladder are manually measured, the operator must calculate the volume of the bladder by multiplying these three values by a correction coefficient. The original correction coefficient can be derived from the formula for calculating hypothetical bladder volume to be a spherical or ellipsoid shape; that is, 4/3 π × (width/2) × (depth/2) × (height/2), which can be simplified as width × depth × height × 0.52 [15]. Because of the underestimation of bladder volume, various correction coefficients other than 0.52 have been recommended in different studies, particularly in cases involving large amounts of PVRU [16]. Variations in bladder shape may result in errors in the ultrasonic estimation of the bladder volume. Bih et al. [11] recommended an optimal correction coefficient of 0.72 for the entire data set and optimal correction coefficients of 0.66, 0.81, and 0.89 for triangular, ellipsoid, and cuboidal bladder shapes, respectively. Cho et al. [17] developed a deep-learning measurement system for bladder volume for use in point-of-care settings. To achieve high accuracy, they used a segmentation model based on a lightweight convolutional neural network. They subsequently used this integrated system to evaluate bladder volume with different coefficients, and they calculated with an error of less than 10% with appropriate shape coefficients, namely 0.52 for spherical bladder shapes, 0.66 for triangular shapes, 0.72 for unknown shapes, 0.81 for cylinder shapes, and 0.89 for cuboid shapes. Overall, they indicated that a coefficient of 0.72 provided the most accurate measurement for both 50- and 150-mL phantoms. Hence, in accordance with these studies, we selected similar correction coefficients to compare their correlations with PVRU volumes instead of bladder shape.

Automated 3D scanning ultrasound provides a rapid and convenient measurement of PVRU volume by automatically scanning the urinary bladder without direct visualization. Multiple studies have indicated the feasibility and the effectiveness of automated 3D scanning ultrasound for PVRU measurement [18,19]. However, when 3D scanning ultrasound is used, abdominal and pelvic fluid accumulation can confound the results (referred to as Type II deviation in this study) [20,21]. Another potential confounding factor in 3D scanning ultrasound lies in the accuracy of the automated tracking function used for depicting the outline of the bladder (referred to as Type I deviation in this study; Figure 3). In our study, Device B provides the pre-scanning images depicting the 2D structure of the bladder and 12 post-scanning images with automated tracking lines in six frames to enable tracking line adjustment. However, examining all 12 images frame by frame is time-consuming and may necessitate another scan if an incorrect target organ is scanned. In our real-world scenario of routine clinical care, none of the ultrasound operators examined the post-scanning images to ensure the accuracy of the automated tracking results, because the first frame containing the first two images typically displayed a recognizable bladder with an acceptable automated tracking line. Brouwer et al. [22] discovered that the pre-scanning option for automated 3D scanning ultrasound did not increase the accuracy of bladder volume estimation. They recommended that the algorithms used for measuring bladder volume be improved. In our study, we discovered that up to 40% of the measurement deviations observed occurred with Device B. Therefore, improving the capability of bladder recognition and the automated tracking function of the bladder wall is necessary to reduce measurement deviations.

Some discrepancies have been observed between real-time B-mode ultrasound and automated 3D scanning ultrasound in terms of the accuracy of PVRU volume estimation. In this study, we reported a strong linear correlation (*r* = 0.873) between PVRU volumes measured by two models of POCUS devices. However, we detected only a moderate level of agreement between the models (ICC = 0.688). These findings are consistent with a report indicating a strong correlation but poor clinical agreement between these two devices [23]. Prentice et al. [24] compared the performance of real-time B-mode ultrasound and automated 3D scanning ultrasound in an intensive care unit and discovered that real-time ultrasound was more accurate. Matsumoto et al. [25] compared five hand-held ultrasound devices and discovered that real-time B-mode ultrasound with a manual diameter method exhibited higher validity for measuring small amounts of urine compared with automated 3D scanning ultrasound with an area method. Dudley et al. [23] reported that real-time ultrasound imaging, when used correctly, provided more accurate results than those provided by automated 3D scanning ultrasound. These findings demonstrate that bladder visualization generally provides more accurate results.

Because of the small sample size (only six samples of urine measured with urinary catheterization), we were unable to compare the PVRU volumes measured with ultrasound and catheterization. We discovered that the PVRU volumes measured with both real-time B-mode ultrasound and automated 3D scanning ultrasound were lower than the volumes measured with urinary catheterization, which may partly be due to a delay in catheterization. Generally, catheterization delays of 30 min after a bladder ultrasound examination are common in the clinical settings of inpatient departments. Real-time B-mode ultrasound has been reported to underestimate the bladder volumes when a correction coefficient of 0.52 is used, especially in cases involving larger amounts of PVRU [26]. Therefore, different coefficients should be used, depending on the shape of the bladder, to estimate bladder volume. However, classifying bladder shape and memorizing appropriate coefficients are both difficult and impractical for calculating bladder volume during each ultrasound examination. Nagle et al. [27] reported that automated 3D scanning ultrasound with manual tracing of the bladder outline provided the most precise nonuniform bladder geometry. However, compared with the 2D diameter method, manually tracing the bladder outline on 12 images is more time-consuming. If the high incidence of both Type I and II measurement deviations is corrected, automated 3D scanning ultrasound may become an optimal method for estimating bladder volumes. This scenario is similar to the use of Device B without measurement deviations. Large PVRU volumes were often associated with ellipsoid and cuboid bladders (Table 4). The agreement between Devices A and B increased with large PVRU volumes and large correction coefficients (Table 5). Therefore, on the basis of these findings, a more convenient and precise real-time B-mode ultrasound device with an appropriate correction coefficient, depending on PVRU volume instead of bladder shape, could be developed. If the measured bladder volume is <100 mL, the device could output the PVRU volume with a built-in correction coefficient of 0.52. If the measured volume is 100–200 mL at a coefficient of 0.52, namely 150 mL, the device could automatically convert this coefficient into 0.66, that is, by multiplying the measured volume by 1.3 (0.66/0.52 = 1.3), and output a final result of 195 mL. Similarly, if the measured volume is >200 mL, the coefficient could be converted into 0.81, that is, by multiplying the measured volume by 1.6 (0.81/0.52). These adjustments could simultaneously address the shortcomings of large measurement deviations in automated 3D scanning ultrasound and the underestimation of PVRU volume in real-time B-mode ultrasound.

Most of the operators in our study preferred automated 3D scanning ultrasound because of its easy operation and learning process. Automated 3D scanning ultrasound requires a short time to complete an ultrasound scan and is optimal for cases that simply require PVRU volume evaluations, provided that measurement deviations can be reduced to an acceptable range. Real-time B-mode ultrasound offers multipurpose clinical applications not limited to the urinary bladder. Although this technique requires a longer time to learn and operate compared with automated 3D scanning ultrasound, its lower incidence of measurement deviations can aid in avoiding incorrect diagnoses. Its convenience and accuracy can also be greatly improved by adjusting its built-in automated calculation formula with appropriate correction coefficients for different PVRU volumes without requiring bladder shape recognition.

Overall, the greatest benefit of bladder POCUS lies not in its accurate measurement of PVRU volume but rather in its ability to detect UTIs early and in the subsequent clinical decision-making process, such as which bladder training program to implement and whether to provide intermittent or Foley catheterization [4,18]. Based on our findings, we have proposed a standardized clinical pathway for decision-making with respect to different amounts of PVRU volumes. Repeated bladder POCUS to measure PVRU volumes is important for patients with larger amounts of PVRU. In this study, two patients underwent bladder POCUS three times on separate days to ensure an improvement of urinary function after appropriate management. In patients with acute ischemic stroke, a PVRU volume of ≥100 mL is the optimal cutoff for UTI prediction; this optimal cutoff warrants further investigation [4]. The prices of newly developed POCUS devices can be expected to gradually decrease. The price difference of the two models of POCUS devices is not much, and the price of Device A is slightly lower. A previous study has shown that the use of bladder POCUS is beneficial in reducing the incidence of UTI and shortening the length of hospital stay [4]. Balancing cost versus patient safety is not difficult. A further study of the long-term costs of POCUS devices, measurement procedures, and cost savings from reduced UTI and shorter hospital stays could help clarify its benefits.

This study has certain limitations. First, our small sample of individuals with urinary catheterization and catheterization delay limited our ability to conduct comparisons with ultrasound-measured PVRU. In real-world scenarios of inpatient medical care, various management strategies can be used to improve urinary retention without catheterization. Therefore, urgent urinary catheterization may not be required within 30 min of a bladder POCUS scan. Second, the 1-month rotational resident physicians who were responsible for the bladder POCUS examination may not have been proficient in the examination’s techniques, which may have influenced the results. However, we observed no difference in the incidence of measurement deviations between the rotational resident physicians and advanced practice registered nurses. Nevertheless, sufficient training is crucial to maintain quality. Third, the pre-scanning images of Device B with a manually adjusted contour tracking function were not utilized in any measurements. None of the ultrasound operators perceived any anomalies in the scanning results, and therefore, they did not proactively examine whether each image was correctly depicted. If the operators need to carefully examine and correct each depiction error, the total time required to complete a single scan is expected to be much longer than that reported in this study.

## 5. Conclusions

Both real-time B-mode ultrasound and automated 3D scanning ultrasound can be used to effectively measure PVRU volume. Automated 3D scanning ultrasound is convenient for learning and scanning but has large measurement deviations. Real-time B-mode ultrasound accurately visualizes the urinary bladder but tends to underestimate it when the volume of PVRU is large. Hence, real-time B-mode ultrasound with automated PVRU-based adjustment of calculation formulas may be a better solution for estimating bladder volume.

## Figures and Tables

**Figure 1 diagnostics-13-02599-f001:**
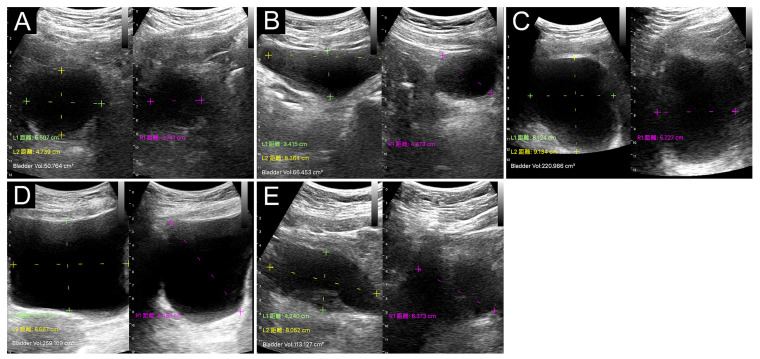
Urinary bladder shapes identified with Device A with split-image mode (left: horizontal plane; right: vertical plane): (**A**) spherical shape, (**B**) triangular shape, (**C**) ellipsoid shape, (**D**) cuboid shape, and (**E**) undefined shape. The values of measured distance (green and yellow dashed lines on the horizontal plane and pink dashed line on the vertical plane) and automatically calculated volume are displayed on the lower left corner of each image.

**Figure 2 diagnostics-13-02599-f002:**
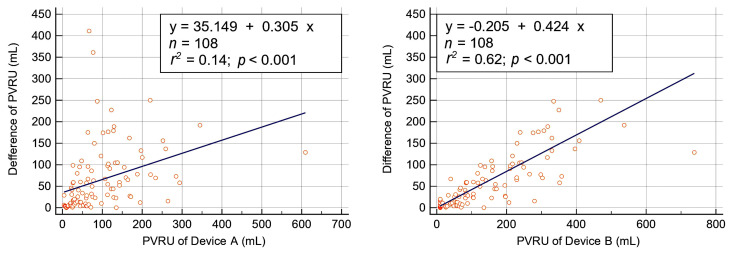
Linear regression analysis of the difference between post-void residual urine (PVRU) volumes measured using Device A (**left**) and Device B (**right**).

**Figure 3 diagnostics-13-02599-f003:**
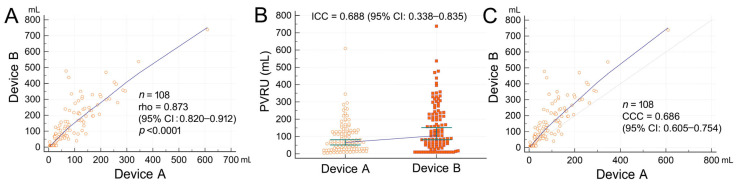
(**A**) Spearman’s rank correlation coefficient analysis indicating a strong positive linear correlation between the post-void residual urine (PVRU) volumes measured using Devices A and B. (**B**,**C**) Intraclass correlation coefficient (ICC) and concordance correlation coefficient (CCC) revealing a moderate agreement between PVRU volumes measured using Devices A and B. CI, confidence interval.

**Figure 4 diagnostics-13-02599-f004:**
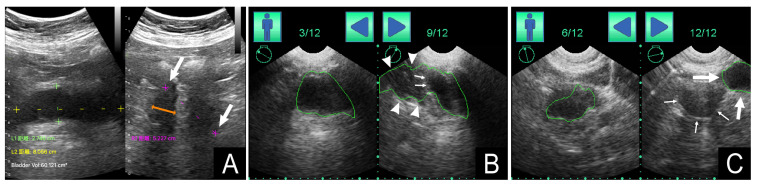
(**A**) The large white arrows indicate a Type I measurement deviation in Device A for measuring bladder height (pink dotted line). The orange arrow indicates the correct distance of measurement. (**B**) The white arrowheads indicate a Type I measurement deviation in Device B for automatically tracking the contour of the bladder wall (green tracking line). The small white arrows indicate the correct outline of the bladder. (**C**) The large arrows indicate a Type II measurement deviation in Device B, in which a hypoechoic area (green tracking line) is mistaken for the bladder (the small arrows indicate the urinary bladder).

**Figure 5 diagnostics-13-02599-f005:**
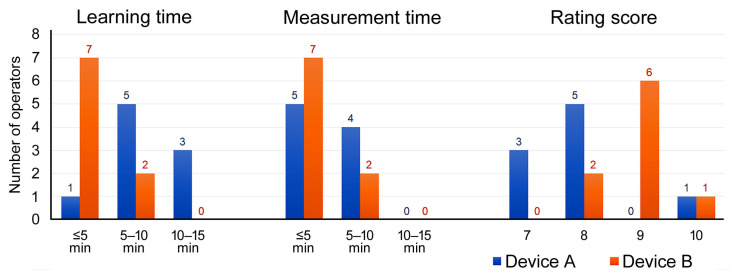
Questionnaire results of the nine ultrasound operators.

**Table 1 diagnostics-13-02599-t001:** Basic characteristics of the 55 participants.

Characteristics	Total (*n* = 55)	Men (*n* = 23)	Women (*n* = 32)	*p* Value
Age (years)	74 (65–85)	80 (70–84)	71 (63–85)	0.194
Body mass index	23.4 (21.4–26.7)	24.7 (22.8–27.3)	23.2 (20.9–24.9)	0.031
AGN3 score	2 (1–2)	1 (1–2)	2 (1–3)	0.102
Number of POCUS measurements	108	44	64	
TOAST classification				
Small artery occlusion	28	10	18	
Large artery atherosclerosis	16	9	8	
Cardioembolism	9	4	4	
Other determined etiology	2	0	2	

AGN3: age ≥ 75 years, female gender, initial National Institutes of Health Stroke Scale (NIHSS) score ≥ 5, NIHSS conscious score ≥ 1, and NIHSS leg score ≥ 2; POCUS, point-of-care-ultrasound; TOAST, trial of ORG 10,172 in acute stroke treatment.

**Table 2 diagnostics-13-02599-t002:** Results of PVRU volumes measured by bladder ultrasound.

Device	Total (*n* = 108)	Men (*n* = 43)	Women (*n* = 65)	*p* Value ^1^
Device A (mL)	65 (28–129)	113 (46–158)	52 (26–105)	0.013
Device B (mL)	105 (42–230)	159 (82–237)	73 (26–220)	0.042
*p* value ^2^	<0.001	<0.001	<0.001	
Device A PVRU > 100 mL	40 (37%)	23 (53%)	17 (26%)	0.005
Device B PVRU > 100 mL	56 (52%)	28 (65%)	28 (43%)	0.031

^1^ Mann-Whitney U test; ^2^ Wilcoxon test; PVRU, post-void residual urine.

**Table 3 diagnostics-13-02599-t003:** Measurement deviations of bladder ultrasound for PVRU volume detection.

	Device A			Device B		
Characteristics	Deviations (*n* = 11; 10%)	No Deviations (*n* = 97; 90%)	*p* Value	Deviations (*n* = 43; 40%)	No Deviations (*n* = 65; 60%)	*p* Value
Age (years)	83 (73–87)	71 (65–84)	0.099	75 (66–84)	72 (64–85)	0.702
Female sex	5 (45%)	60 (62%)	0.339	25 (58%)	40 (62%)	0.841
BMI	27.3 (22.8–27.6)	23.3 (21.2–26.1)	0.101	24.1 (21.7–27.3)	23.2 (21.1–24.9)	0.139
PVRU (mL)	97 (65–133)	64 (26–129)	0.089	105 (71–215)	89 (10–253)	0.344

BMI, body mass index; PVRU, post-void residual urine.

**Table 4 diagnostics-13-02599-t004:** Correlation of bladder shape with PVRU volumes and measurement deviations.

	PVRU Volume (mL)		Measurement Deviations		Gender	
Bladder Shape	Device A (*n* = 108)	Device B (*n* = 108)	Device A (*n* = 11)	Device B (*n* = 43)	Women (*n* = 64)	Men (*n* = 44)
Spherical (*n* = 32)	26 (12–46)	26 (10–67)	1 (3%)	13 (41%)	23 (72%)	9 (28%)
Triangular (*n* = 24)	43 (26–74)	71 (46–105)	2 (8%)	9 (38%)	16 (67%)	8 (33%)
Undefined (*n* = 11)	61 (40–91)	89 (49–152)	4 (36%)	6 (55%)	6 (55%)	5 (45%)
Ellipsoid (*n* = 18)	135 (114–159)	204 (141–276)	3 (17%)	7 (39%)	5 (28%)	13 (72%)
Cuboid (*n* = 23)	188 (124–257)	317 (244–386)	1 (4%)	8 (35%)	15 (65%)	8 (35%)
*p* value	<0.001 *	<0.001 *	0.019 **	0.860 **	0.033 **	

* Kruskal-Wallis test; ** chi-square test; data are presented as median (25th–75th percentile) or *n* (%). PVRU, post-void residual urine.

**Table 5 diagnostics-13-02599-t005:** Correlations of PVRU volumes between Device A and Device B at different correction coefficients.

		PVRU (mL)		Difference (%)		Agreement	
		Mean	Median	Mean	Median	ICC	CCC
Total measurements (***n*** = 60)							
Device B		135 ± 135	68 (10–237)				
Device A							
	0.52	89 ± 87	61 (21–135)	34 ± 22	30 (17–52)	0.762	0.759
	0.66	113 ± 110	78 (27–171)	29 ± 23	23 (9–48)	0.872	0.870
	** *0.72* **	** *124* ** ** * ± * ** ** *121* **	** *85 (29–187)* **	** *30* ** ** * ± * ** ** *23* **	** *25 (10–47)* **	** *0.902* **	** *0.901* **
	0.81	139 ± 136	95 (33–210)	32 ± 21	29 (15–49)	0.894	0.892
Measurement with PVRU < 100 mL (*n* = 31)							
Device B		27 ± 22	10 (10–47)				
Device A							
	** *0.52* **	** *26* ** ** * ± * ** ** *21* **	** *22 (12–30)* **	** *33* ** ** * ± * ** ** *23* **	** *29 (12–56)* **	** *0.545* **	** *0.537* **
	0.66	33 ± 26	28 (15–38)	34 ± 24	29 (11–57)	0.528	0.519
	0.72	36 ± 28	30 (16–42)	36 ± 24	33 (18–54)	0.505	0.497
	0.81	40 ± 32	34 (18–47)	42 ± 21	38 (27–54)	0.464	0.458
Measurement with PVRU 100–200 mL (*n* = 12)							
Device B		158 ± 160	160 (132–189)				
Device A							
	0.52	116 ± 41	127 (73–141)	27 ± 18	26 (14–38)	0.460	0.437
	** *0.66* **	** *148* ** ** * ± 52* **	** *162 (93–179)* **	** *17* ** ** * ± * ** ** *15* **	** *10 (7–23)* **	** *0.691* **	** *0.671* **
	0.72	161 ± 56	176 (101–195)	17 ± 14	16 (3–25)	0.683	0.662
	0.81	181 ± 63	198 (114–220)	21 ± 11	24 (13–26)	0.579	0.555
Measurement with PVRU > 200 mL (*n* = 17)							
Device B		317 ± 77	318 (262–351)				
Device A							
	0.52	188 ± 83	188 (120–255)	41 ± 21	38 (25–61)	0.297	0.282
	0.66	239 ± 105	238 (153–324)	29 ± 22	22 (9–50)	0.479	0.308
	0.72	261 ± 115	260 (167–333)	26 ± 21	17 (11–46)	0.543	0.342
	** *0.81* **	** *293* ** ** * ± * ** ** *129* **	** *293 (188–397)* **	** *24* ** ** * ± * ** ** *19* **	** *17 (9–39)* **	** *0.591* **	** *0.576* **

CCC, concordance correlation coefficient; ICC, intraclass correlation coefficient; PVRU, post-void residual urine; values displayed in bold and italics exhibit the strongest agreement with Device B.

## Data Availability

The data presented in this study are available on request from the corresponding author.

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
