# Peer review of "Comparative Effectiveness of Two Models of Point-of-Care Ultrasound for Detection of Post-Void Residual Urine during Acute Ischemic Stroke: Preliminary Findings of Real-World Clinical Application"

_diagnostics, 2023, doi:10.3390/diagnostics13152599_

Round 1

Reviewer 1 Report

I would like to congratulate the authors on the selection of this novel and the interesting aspect of cardiology. The authors have provided good evidence to support their conclusion with well constructed and meticulously written manuscript. 

However, I would like to bring attention to the following points

1. Please elaborate statistical section on the methodology 

2. In the results section provided results in graphical format can be helpful for readers 

3. In the limitations section more succinct presentation is needed. 

The English language needs minor revisions otherwise is in acceptable format

Author Response

Manuscript ID diagnostics-2531662

Title: Comparative effectiveness of two models of point-of-care ultrasound for detection of post-void residual urine during acute ischemic stroke: real-world clinical application

Thanks to reviewer’s precious comments. We have checked the manuscript and have made essential revisions according to reviewer’s comments point-by-point. We have also corrected some typos in the text. The revised portions in the manuscript were coded in red color.

Comments from Reviewer 1:

I would like to congratulate the authors on the selection of this novel and the interesting aspect of cardiology. The authors have provided good evidence to support their conclusion with well constructed and meticulously written manuscript. 

However, I would like to bring attention to the following points

  1. Please elaborate statistical section on the methodology 

Response: We have added more statistical methods in the methods section which were missed before but were mentioned in the results section.

  1. In the results section provided results in graphical format can be helpful for readers.

Response: We have changed Table 6 with graphical format into Figure 5 to present the questionnaire results of the nine ultrasound operators.

  1. In the limitations section more succinct presentation is needed.

Response: We have deleted some sentences in limitations section to make it more succinctly.

Reviewer 2 Report

The authors conducted a prospective comparative clinical study aimed at better understanding the effectiveness and convenience of two models of bladder point-of-care ultrasound (POCUS) devices for rapid bedside measurements of post-void residual urine (PVRU) volume in a sample of 55 acute ischemic stroke patients. The authors found that both real-time B-mode ultrasound and automated 3D scanning ultrasound can be used to effectively measure PVRU volume. However, real-time B-mode ultrasound with automated  PVRU-based adjustment of the calculation formulas may a better solution to estimate bladder volume. The study is potentially  interesting, but can be improved if the following considerations are addressed:     

1)Due to the small size of the study, the title should clearly mention “preliminary findings”

2) It would be interesting to know the different stroke subtypes in the study population.

3) The topography of cerebral ischemia should be clarified 

4) It may be also of interest to determine whether "urinary events" appear as an independent significant predictor of prolonged hospital stay in acute stroke patients as observed by other authors (add a comment of a related study published in (International Journal of Clinical Medicine 2012; doi: DOI: 10.4236/ijcm.2012.36090).   

5) The authors should indicate that an essential line of future research would be precisely the assessment of the comparative effectiveness of the two models of point-of-care ultrasound for the detection of post-void residual urine in patients with lacunar versus non-lacunar ischemic stroke. This recommendation is because the pathophysiology, prognosis, and clinical features of lacunar strokes are different from other acute cerebrovascular diseases (Int J Mol Sci 2022; 23, 1497). Did the authors consider this in their study protocol?

Author Response

Manuscript ID diagnostics-2531662

Title: Comparative effectiveness of two models of point-of-care ultrasound for detection of post-void residual urine during acute ischemic stroke: real-world clinical application

Thanks to reviewer’s precious comments. We have checked the manuscript and have made essential revisions according to reviewer’s comments point-by-point. We have also corrected some typos in the text. The revised portions in the manuscript were coded in red color.

Comments and Suggestions for Authors from Reviewer 2:

The study is potentially interesting, but can be improved if the following considerations are addressed: 

1) Due to the small size of the study, the title should clearly mention “preliminary findings”

Response: We have changed the title to “preliminary findings” of real-world clinical application.

2) It would be interesting to know the different stroke subtypes in the study population.

Response: We have added the stroke subtypes according to the TOAST classification in Table 1 and addressed the results in the text (line164-166).

3) The topography of cerebral ischemia should be clarified 

 Response: We have added the distribution of infarct location in the results (line 166 – 169).

4) It may be also of interest to determine whether "urinary events" appear as an independent significant predictor of prolonged hospital stay in acute stroke patients as observed by other authors (add a comment of a related study published in (International Journal of Clinical Medicine 2012; doi: DOI: 10.4236/ijcm.2012.36090).   

 Response: We have discussed more about the influence of urinary events on the prolonged hospital stay in [Introduction] (line 41-42) and have added the recommended article as a reference (Ref. 1).

5) The authors should indicate that an essential line of future research would be precisely the assessment of the comparative effectiveness of the two models of point-of-care ultrasound for the detection of post-void residual urine in patients with lacunar versus non-lacunar ischemic stroke. This recommendation is because the pathophysiology, prognosis, and clinical features of lacunar strokes are different from other acute cerebrovascular diseases (Int J Mol Sci 2022; 23, 1497). Did the authors consider this in their study protocol?

Response: This study focuses on the comparison of the detected PVRU volumes between two models of POCUS devices. Because of the small sample size, we did not stratify patients into lacunar and non-lacunar stroke. Thanks to reviewer’s recommendation. It would be interesting to compare the PVRU volumes among different subtypes of stroke. The bladder POCUS procedure has become a routine procedure at our stroke ward. We will analyze the difference during our long-term bladder POCUS study.

Reviewer 3 Report

Point-of-care ultrasound (POCUS) is a portable and non-invasive imaging technique that allows healthcare professionals to visualize and assess various structures in real-time. It is often used at the bedside or in the emergency setting to obtain immediate information and aid in making clinical decisions.

One aspect of stroke management involves monitoring post-void residual urine (PVRU) in stroke patients to monitor to prevent urinary retention and potential complications, especially in patients with mobility issues. POCUS can be used to assess PVRU in stroke patients as part of their routine neurological examination. By visualizing the bladder and measuring the volume of residual urine, healthcare professionals can promptly address any urinary retention issues and provide appropriate care to the patient.

For a study to compare the effectiveness of two different models of POCUS for detecting PVRU during acute ischemic stroke, it would likely involve evaluating the accuracy, reliability, and ease of use of each model in a real-world clinical setting.

It was an innovative and interesting study in the follow-up of patients who have suffered a stroke. While the authors describe the use and clinical advantages of different models, it would have been interesting to include possible costs and clinical impacts of decision changes in patient care based on the device used. Does the improved precision in measuring PVRU with the automated device have an impact on subsequent clinical decisions?

Author Response

Manuscript ID diagnostics-2531662

Title: Comparative effectiveness of two models of point-of-care ultrasound for detection of post-void residual urine during acute ischemic stroke: real-world clinical application

Thanks to reviewer’s precious comments. We have checked the manuscript and have made essential revisions according to reviewer’s comments point-by-point. The revised portions in the manuscript were coded in red color.

Comments and Suggestions for Authors from Reviewer 3:

It was an innovative and interesting study in the follow-up of patients who have suffered a stroke. While the authors describe the use and clinical advantages of different models, it would have been interesting to include possible costs and clinical impacts of decision changes in patient care based on the device used. Does the improved precision in measuring PVRU with the automated device have an impact on subsequent clinical decisions?

Response: We have added more descriptions about the impact of bladder POCUS on the decision-making and the possible costs in the text (line 420-433). We have also corrected some typos in the text.
